# Complex Temperature and Concentration Dependent Self-Assembly of Poly(2-oxazoline) Block Copolymers

**DOI:** 10.3390/polym12071495

**Published:** 2020-07-04

**Authors:** Loan Trinh Che, Marianne Hiorth, Richard Hoogenboom, Anna-Lena Kjøniksen

**Affiliations:** 1Department of Chemistry, University of Oslo, P.O. Box 1033 Blindern, 0315 Oslo, Norway; loan.trinh.che@gmail.com; 2Department of Pharmacy, School of Pharmacy, University of Oslo, P.O. Box 1068 Blindern, 0316 Oslo, Norway; marianne.hiorth@farmasi.uio.no; 3Supramolecular Chemistry Group, Centre of Macromolecular Chemistry (CMaC), Department of Organic and Macromolecular Chemistry, Ghent University, Krijgslaan 281 S4, B-9000 Ghent, Belgium; 4Faculty of Engineering, Østfold University College, P.O. Box 700, 1757 Halden, Norway

**Keywords:** responsive polymers, lower critical solution temperature, poly(2-oxazoline)s, block copolymer

## Abstract

The effect of polymer concentration on the temperature-induced self-association of a block copolymer comprising a poly(2-ethyl-2-oxazoline) block and a random copolymer block consisting of 2-ethyl-2-oxazoline and 2-*n*-propyl-2-oxazoline (PEtO_80_-*block*-P(EtOx_x_-*stat*-PropO_40-x_) with x = 0, 4, or 8 were investigated by dynamic light scattering (DLS) and transmittance measurements (turbidimetry). The polymers reveal a complex aggregation behavior with up to three relaxation modes in the DLS data and with a transmittance that first goes through a minimum before it declines at high temperatures. At low temperatures, unassociated polymer chains were found to co-exist with larger aggregates. As the temperature is increased, enhanced association and contraction of the aggregates results in a drop of the transmittance values. The aggregates fragment into smaller micellar-like clusters when the temperature is raised further, causing the samples to become optically clear again. At high temperatures, the polymers aggregate into large compact clusters, and the samples become turbid. Interestingly, very large aggregates were observed at low temperatures when the polymer concentrations were low. The formation of these aggregates was also promoted by a more hydrophilic copolymer structure. The formation of large aggregates with an open structure at conditions where the solvent conditions are improved is probably caused by depletion flocculation of the smaller aggregates.

## 1. Introduction

Synthetic amphiphilic macromolecules are able to form self-assembled structures in aqueous media. These polymers are well known as emulsifiers and viscosity modifiers for industrial applications [1,2,3,4,5]. Stimuli-responsive amphiphilic copolymers are “smart” materials that have the ability to respond to external stimuli such as temperature, pH, electric or magnetic fields, light, and mechanical stress. Among them, temperature and pH responsive mechanisms have been considerably investigated during the past decades due to their potential for biomedical applications such as bio-sensors, implants, and drug-delivery devices [6]. Alteration of molecular interactions as a result of temperature changes provides systems that are useful in bio-related applications [7,8,9,10,11,12]. Thermoresponsive polymers in water may exhibit a lower critical solution temperature (LCST), i.e., an increased hydrophobicity at elevated temperatures, or an upper critical solution temperature (UCST) where the polymers exhibit enhanced associations when the temperature is decreased [13,14,15,16,17]. Many amphiphilic polymers undergo a micellization/demicellization by the alternation of the balance between hydrophilicity and hydrophobicity due to stimuli-induced collapse of one of the blocks. This has been used for the encapsulation of insoluble drugs [18], which can be released to a target area of the body. The hydrophobic blocks at the cores of the particles provide a suitable location for accommodation of poorly soluble drugs. The hydrophilic coronas stabilize the nanoparticles in aqueous environments. Many different thermoresponsive polymers such as poly(*N*-isopropylacrylamide) (PNIPAAM) [14,19,20], pol(oligoethyleneoxide (meth)acrylate)s [21,22,23], and poly(2-oxazoline)s [18,24,25,26,27,28] have been tested over the years for temperature-induced micellization. Among these polymers, poly(ethylene glycol) (PEG) is the most exploited polymer in the field of drug delivery and nanoparticle biocompatibilization [29], because of its low toxicity, high solubility, and non-immunogenicity [29,30,31,32,33] as well as its Food and Drug Administration (FDA) approval in pharmaceutical formulations. However, PEG also has some undesirable features, such as interactions with various immunological entities and redundant accumulation in the body [29]. Therefore, poly(2-alkyl-2-oxazoline)s (PAOx) polymers are proposed as promising alternatives to PEG, and more attention is directed towards the use of PAOx in biomedical applications [34,35,36].

PAOx are often used in adhesives [37,38,39], coatings [40] or ink formulations [41], as well as in drug delivery applications [18,34,35,36]. They are especially popular in the design of drug nanocarriers [42]. Studies of PAOx were already established more than 50 years ago [43,44,45,46,47]. However, during the last few decades, the interest in PAOx has increased considerably. The discovery of their potential use as biomaterials and thermoresponsive systems has led to an exponential growth in patent registration of the PAOx polymers in the last decade [35]. In addition, poly(2-ethyl-2-oxazoline) (PEtOx) has been approved by the FDA as an indirect additive used in food contact substances [25], and a PAOx based therapeutic has successfully completed Phase 1A clinical trials [48].

PAOx contains C–C–N as the repeating units of the backbone and amide side chains, and are therefore described as pseudo-polypeptides because of their similar isomeric structure [49]. Furthermore, PAOx polymers exhibit high functionalization possibilities and versatile chemistry [50]. Due to the versatility of PAOx, chemical functionalities can be introduced into the PAOx structure, either within the polymer backbone or at both chain ends. Accordingly, the polymers can be designed to achieve different architectures. This enables tailor-made physico-chemical properties such as thermal transitions, mechanical properties, solubility and surface energy [51,52].

In aqueous solutions, the water affinity of PAOx polymers depends on the type of alkyl side chain that is present. Poly(2-methyl-2-oxazoline) (PMeOx) is water-soluble in the entire temperature range of liquid water under atmospheric pressure. Poly(2-ethyl-2-oxazoline) (PEtOx) and poly(2-isopropyl-2-oxazoline) (PiPropOx) exhibit a LCST behavior in aqueous solution, whereas longer substituents result in hydrophobic polymers [24]. In the case of hydrophilic water-soluble polymers used for drug delivery, PMeOx and PEtOx are often compared to PEG due to similar hydrophilicity [16,53], biocompatibility [54], blood circulation times [53,55], and stealth behavior [16,53,55]. Moreover, it was found that PEtOx is more lipophilic than PMeOx and PEG, which can be advantageous for loading non-soluble drugs in a delivery system [16,56]. In addition, Viegas et al. reported that linear and branched PEtOx have a significantly lower viscosity than PEG, especially at higher molecular weights. This observation is important because it allows better pharmaceutical formulations, filterability, and syringeability of PEtOx in protein and peptides conjugations at higher concentrations [16].

Combining a hydrophilic PAOx block with a different hydrophobic PAOx block or other hydrophobic polymers is a straightforward way for the preparation of amphiphilic PAOx block copolymers. Aqueous solutions of PEtOx have a LCST in the range of 61–64 °C [57], while poly(2- PiPropOx exhibits a cloud point around 36–39 °C [58]. The latter transition temperature is very close to body temperature and therefore has promising possibilities for biomedical applications. Living cationic copolymerization of different types of 2-oxazoline monomers in a sequential manner yields well-defined, amphiphilic, block copolymers [24]. Accordingly, the LCST behavior can be tuned by varying the composition and the degree of polymerization in random PAOx copolymers to prepare thermoresponsive materials [59,60]. Kataoka et al. prepared a series of poly(2-*n*-propyl-2-oxazoline) (PnPropOx) mixed with a specific composition of EtOx, and reported that the LCST of the copolymers can be modulated over a broad range of temperature from 39 to 67 °C [24].

We have previously studied the behavior of PEtOx_80_-*block*-P(EtOx_x_-*stat*-PropOx_40-x_) with x = 0, 4 or 8 at a fixed concentration of 5 mg/mL in water [61]. The block copolymers respond to temperature changes and revealed a complex aggregation behavior that depends on the difference in hydrophilicity between the more hydrophilic PEtOx block and the PEtOx-*stat*-PropOx block. In order to gain a deeper understanding of the associating mechanism of these systems in dilute solution, a more detailed investigation on the effect of block copolymer concentration on the temperature-induced micellization behavior is carried out in this work by dynamic light scattering (DLS) and turbidimetry measurements. By studying the aggregation behavior of PEtOx-*block*-P(EtOx-*stat*-PropOx) in aqueous solution, we aim at developing a deeper insight into the factors that govern the dynamics of self-organization of this type of systems.

## 2. Materials and Methods

The synthesis and characterization of the studied block copolymers has been reported in our previous work [61]. The block copolymers consist of a more hydrophilic block of PEtOx with a degree of polymerization of 80 and a second more hydrophobic block consisting of PEtOx-*stat*-PropOx with a degree of polymerization of PEtOx_80_-*block*-PPropOx_40_ has *M*_n_ = 8700 g/mol and PDI = 1.22; PEtOx_80_-*block*-P(EtOx_4_-*stat*-PropOx_36_) has M_n_ = 8200 g/mol and PDI = 1.23; PEtOx_80_-*block*-P(EtOx_8_-*stat*-PropOx_32_) has M_n_ = 9200 g/mol and PDI = 1.20 [61]. Solutions of the polymers were prepared in deionized water, and were stirred at room temperature until all polymer was dissolved or dispersed.

Transmittance measurements: The transmittance of the polymer solutions (5 mg/mL) were determined by turbidity measurements using a Crystal 16 (Avantium Technologies, The Netherlands) connected to a Julabo FP40 cryostat. Turbidity of the solutions was measured by the transmission of red light through the sample vial as a function of the temperature while stirring. Solutions of the polymers were prepared in deionized water (Laborpure, Behr Labor Technik) and were stirred at room temperature until all polymer was dissolved or dispersed. The samples were measured from 20 to 80 °C with a heating rate of 1 °C/min.

Dynamic light scattering: The dynamic light scattering (DLS) experiments were performed with an ALV/CGS-8F goniometer system, with 8 fiber-optical detection units, from ALV-GmbH, Langen, Germany. Solutions of the polymers were prepared in deionized water (Laborpure, Behr Labor Technik) and were stirred at room temperature until all polymer was dissolved or dispersed Different sample solutions with altered concentrations (1, 2.5, 5, and 7.5 mg/mL) were filtered in an atmosphere of filtered air through a 5 μm filter (Millipore) directly into pre-cleaned 10 mm NMR tubes (Wilmad Glass Co. Vineland, NJ, USA) of highest quality. The samples were measured from 25 to 60 °C with a heating rate of 1 °C/min.

The DLS experiments probe the concentration fluctuations relaxation toward equilibrium at a length scale 1/q. The magnitude of the wave vector, q, is given by q = (4πn/λ)sin(θ/2), where λ is the wavelength of the incident light in a vacuum, θ is the scattering angle, and n is the refractive index of the medium. Assuming that the scattering of the incoming light exhibits Gaussian statistics, the experimentally recorded intensity autocorrelation function g^2^(q,t) is directly linked to the theoretically amenable first-order electric field autocorrelation function g^1^(q,t) through the Siegert [62] relationship: g^2^(q,t) = 1 + B|g^1^(q,t)|^2^, where B (≤1) is an empirical factor that depends on the experimental geometry.

Depending on the conditions, the correlation functions were found to exhibit either one, two or three relaxation modes. In this DLS study, the correlation functions that only showed a simple exponential decay were fitted by
g^1^(q,t) = exp[−(t/τ_se_)^β^](1)
where τ_se_ represent an effective relaxation time and β express the width of the distribution of the relaxation times. The value of β is limited to the interval 0 < (β) ≤ 1.

In cases that are more complex it is often observed that the decay of the correlation functions exhibits two relaxation modes. The correlation functions were then fitted by the sum of a single and a stretched exponential:g^1^(q,t) = A_f_ exp(−t/τ_f_) + A_s_ exp[−(t/τ_se_)^β^](2)
where A_f_ + A_s_ = The parameters A_f_ and A_s_ are the amplitudes while τ_f_ and τ_se_ are the relaxation times for the fast and the slow relaxation modes, respectively.

In the analysis of the recent work, the correlation functions were also found to exhibit an additional third mode at very long times (Figure 1). The correlation function can be fitted by
g^1^(q,t) = A_f_ exp(−t/τ_f_) + A_s_ exp[− (t/τ_se_)^β^] + A_vs_ exp(−t/τ_vs_)(3)
where A_f_ + A_s_ + A_vs_ = The subscript f is associated with the fast mode, subscript s relates to the slow mode, and the subscript vs is connected to the very slow mode. The mean relaxation time is given by $\tau_{s}=\frac{\tau_{se}}{\beta }\Gamma \left(\frac{1}{\beta }\right)$, where Γ is the gamma function. It was found that the stretched exponent of the fast mode and the very slow mode were very close to one, and therefore a single exponential was used in order to reduce the number of fitting parameters.

The third mode in the relaxation process were observed at the lowest concentration for all the investigated polymers. In the case of PEtOx_80_-*block*-P(EtOx_8_-*stat*-PropOx_32_) (the most hydrophilic of these polymers), this feature also appeared at the concentration of 2.5 mg/mL.

In order to determine the number of relaxation modes for each correlation function, residual plots have been employed. A plot of the residuals should always be used in order to check how well the analysis fits the data. In this way, the agreement between the methods of data analysis, the parameters received, and the measured data can be checked (Figure 1).

The q dependences of the inverse fast, slow, and very slow relaxation times can be quantified by $\tau_{f}^{-1}\sim q^{\alpha_{f}}$,$\tau_{s}^{-1}\sim q^{\alpha_{s}}$, and $\tau_{vs}^{-1}\sim q^{\alpha_{vs}}$, respectively. The value of α_f_ ≈ 2 at all conditions, except for the correlation functions that exhibit a trimodal distribution. For the cases where α_f_ ≈ 2, it indicates the existence of a diffusive mode, and the Stokes–Einstein relationship can be used to calculate the hydrodynamic radius of the fast relaxation mode:$R_{h,f}=\frac{k_{B}T}{6\pi \eta D_{f}}$, where k_B_ is the Boltzmann constant, T is the absolute temperature, and η is solvent viscosity. The mutual diffusion coefficient of the fast mode can be expressed as D_f_ = 1/(*τ*_f_ q^2^). At low concentrations, in the presence of the trimodal distribution, the value of α_f_ were found to alter between 2 to 4 and α_s_ varied from 2 to At higher concentrations, the fast mode is diffusive, while the slow relaxation mode exhibits stronger q dependence at some temperatures. A q-dependency higher than 2 might be caused by interactions between the particles [63,64] or if qR >> 1 by the influence of internal motions within the particles [65,66,67]. The Stokes–Einstein relationship is not valid when α_s_ > 2, and using it under these conditions will result in different sizes depending on the scattering angle employed in the measurements. However, even though the absolute values will not be correct, the trends in the data should indicate whether the sizes are increasing or decreasing. We have therefore chosen to represent the values calculated from the slow relaxation mode as an apparent hydrodynamic radius, R_h,s_, using a scattering angle of 107°: $R_{h,s}=\frac{k_{B}T}{6\pi \eta D_{s}}$, where the mutual diffusion coefficient of the slow mode is D_s_ = 1/(*τ*_s_ q^2^). The amplitudes of the relaxation modes are angle dependent. The scattering angle of 107° was chosen since it was found to give the best resolution of the modes.

## 3. Results and Discussion

### 3.1. Turbidimetry

The thermoresponsive nature of these copolymers gives rise to a complex turbidity behavior [61]. As can be seen from Figure 2, the transmittance goes through a minimum value followed by an increase, before the transmittance decreases again to very low values. Previous studies on the 5.0 mg/mL samples [61] revealed that there is an aggregation induced decline in the transmittance due to collapse of the PEtOx-PropOx block, followed by fragmentation of the clusters into micellar-like entities causing a rise in the transmittance. At even higher temperatures, the system re-aggregates into large compact clusters due to collapse of the PEtOx block, causing very low transmittance values. Similar transmittance trends have also been observed for other thermoresponsive block copolymers [68,69,70,71,72,73].

A closer look at the transmittance values in Figure 2 reveals an intriguing trend at low temperatures, where the transmittance values do not follow the expected concentration dependency. The transmittance of a sample is dependent on the concentration, the size of the particles, and the difference in refractive index between the particles and the surrounding liquid [74]. The difference in refractive index can be related to how swollen/compact the particles are. A swollen particle contains a high amount of solvent resulting in an overall refractive index that is close to that of the solvent (hence little scattering), while a compact particle has a refractive index closer to that of the polymer leading to more scattering. Provided that the particles are not both very large and very compact, aggregation causes a decrease in the transmittance while swelling results in a higher transmittance [61,75]. If no changes in the aggregation or swelling behavior of the samples take place, the transmittance decreases as the concentration is raised. In addition, increasing the concentration often enhances aggregation [76], which will reduce the transmittance values further. It is therefore surprising to find the trend illustrated in Figure 3a where the transmittance values go through a maximum when the copolymer concentration is raised. This interesting behavior will be discussed in more detail below, in connection with the dynamic light scattering results.

Both higher concentrations and enhanced polymer hydrophobicity (higher amount of PropOx) causes lower transmittance values at the minimum (Figure 2 and Figure 3a,b). In addition, the minimum is shifted towards lower temperatures (Figure 2). This is reasonable since higher concentrations and enhanced hydrophobicity promote both aggregation of the samples and contraction of the aggregates, thereby reducing the transmittance of the samples [61,75].

### 3.2. Dynamic Light Scattering

The dynamic light scattering reveals a complex picture, where the correlation functions exhibit one, two, or even three modes, depending on temperature, type of polymer, and polymer concentration. Great care was taken to ensure that the correct number of modes was used in the fitting procedure (see experimental section for details). Figure 4 shows the hydrodynamic radius (R_h,f_) determined from the fast relaxation mode. This mode appears at low temperatures for all the samples. The small sizes of a few nm suggest that we are probing unassociated unimers (single polymer chains). As expected for these small unimers, there is little size variation between the different polymers and concentrations. When the temperature is raised, R_h,f_ of some of the samples increases, indicating aggregation of the unimers into larger entities. At high temperatures, the unimers are no longer evident in the correlation functions, as enhanced aggregation leaves too few unimers in the solution to be detectable. An increased concentration and higher amount of PropOx (which is more hydrophobic than EtOx) promotes aggregation, and accordingly the temperature where R_h,f_ disappears from the correlation functions is reduced.

The species probed by the second relaxation mode, giving the apparent hydrodynamic radius R_h,s_ (open symbols in Figure 5) are much larger than the unimers. Accordingly, this mode probes large aggregates, illustrating that the copolymers exhibit associative interactions even at temperatures below the LCST. This is not uncommon for thermoresponsive polymers [73,77,78,79], and even double hydrophilic block copolymers have been observed to form aggregates in dilute aqueous solutions [80,81,82]. The latter was explained by differences in hydrophilicities between the two blocks, resulting in different water uptake [80]. It is reasonable to assume that similar effects influence the aggregation behavior of thermoresponsive copolymers below the LCST. The size of the aggregates will increase when the average number of polymers in each aggregate becomes higher. However, the associative forces that promote aggregation also favors contraction of the aggregates, which will reduce the sizes. Accordingly, the overall size of the aggregates is a result of these two competing mechanisms. At most conditions, the size of these aggregates is largest for the highest polymer concentration, which may be expected since a higher concentration promotes enhanced associations of the polymers. However, as is evident from Figure 5b, the lowest concentration does not necessarily have the smallest sizes. This discrepancy is probably caused by a higher degree of swelling of the aggregates for the lower concentrations.

When the temperature is raised, R_h,s_ is initially relatively constant, followed by a size reduction. The decreased sizes are caused by contraction of the aggregates, and are most pronounced for the highest concentration. During this contraction, there is a corresponding slight decrease of the transmittance, while the steep transmittance decline occurs at somewhat higher temperatures (Figure 2). The steep decline of the transmittance is due to the formation of aggregates that are both large and compact. The samples are simultaneously aggregating and contracting. During the first steep decline of the transmittance, most of the samples exhibit an increase of R_h,s_. This indicates that the size increase from aggregation is faster than the size reduction of the contraction at this stage.

Around the same temperature as the transmittance starts to increase again after the minimum (Figure 2), a new mode appears in the correlation functions (R_h,x_; filled symbols in Figure 5). At this stage, the large aggregates (R_h,s_) gradually fragment into smaller micellar-like structures (R_h,x_) [61]. This fragmentation is due to the much higher hydrophobicity of PropOx than EtOx in this temperature region [60]. Accordingly, the copolymers rearrange into micellar-like structures with the hydrophobic block containing PPropOx in the middle and the more hydrophilic PEtOx block pointing out towards the aqueous surroundings [61]. The smaller sizes of R_h,x_ causes the increase in the transmittance, and as more and more of the large aggregates fragment into smaller structures the transmittance continues to rise, and the samples becomes optically clear again. For several of the samples, R_h,x_ becomes smaller when the temperature is raised. This is probably caused by a contraction of the EtOx corona, which becomes more hydrophobic with increasing temperatures, possibly in combination with further dehydration of the PEtOx-PropOx core [61].

At even higher temperatures, the PEtOx block is also passing its LCST phase transition and phase separates from solution. At this stage, the associative forces between the polymers become dominant causing a re-aggregation, and a sharp drop in the transmittance (Figure 2). At these conditions, the samples exhibit multiple scattering, and the DLS data cannot be analyzed. Interestingly, this second drop in the transmittance is shifted towards higher temperatures for the most hydrophobic polymer (PEtOx_80_-*b*-PPropOx_40_) and for the highest concentrations. This is unexpected, since one would expect the aggregation to be promoted by enhanced hydrophobicity and increased concentrations. However, when the hydrophobic interactions of the P(EtOx-*stat*-PropOx) block increases, a higher amount of the polymer clusters will fragment into the micellar-like aggregates (R_h,x_) with a PEtOx outer layer. These structures are stabilized by the PEtOx corona and therefore much less prone to aggregation than the more unstructured R_h,s_ aggregates [61].

The stretched exponent, β, provides information about how broad the size distribution of the aggregates is. β = 1 indicates a monodisperse distribution, and a broader size distribution will give lower values of β. Figure 6 shows β for the slow mode (R_h.s_). The other modes exhibited a nearly monodisperse size distribution, with a stretched exponent very close to one (and the exponent was therefore fixed to one to reduce the number of fitting parameters). A narrow size distribution of R_h,f_ is expected, since the fast mode probes single chains of polymers with a relatively small polydispersity. R_h,x_ is due to micellar-like structures that is also expected to exhibit a narrow size distribution. The β values for R_h.s_ varies between 0.4 and 1, and is generally higher (narrower size distributions) at elevated temperatures where large and compact aggregates are formed. At low temperatures, β decreases when the concentration is raised for the most hydrophobic polymer (PEtOx_80_-*b*-PPropOx_40_), and β is also generally lower than for the other polymers in this temperature range. Accordingly, the aggregates formed at low temperatures have a wider size distribution when there are more hydrophobic interactions in the sample.

At the lowest concentrations, a relaxation mode (τ_vs_) with extremely long relaxation times (large sizes) is observed (Figure 7). Both diluting the system and reducing the hydrophobicity of the statistical PEtOx-PropOx block will result in better solvent conditions for the polymer. Accordingly, the polymer is expected to swell more in the solvent, and have a reduced tendency of forming aggregates. The promotion of a relaxation mode with an extremely slow relaxation time is rather surprising at these conditions. Nonetheless, the decline in the transmittance at low concentrations (Figure 3a) is in accordance with the formation of large aggregates. Although the transmittance decreases for the lowest concentration, it does not become very low. At a constant concentration, the transmittance decreases with the size and compactness of the aggregates [61,75]. Accordingly, very large aggregates with only a moderate reduction of the transmittance indicate that the aggregates have an open structure. This raises the question of what kind of process would promote the formation of large aggregates with an open structure when the solvent conditions are improved.

Elongated or rod-like structures could cause a very slow relaxation mode. However, they should also induce a rotational mode with a q-dependency lower than 2 [83]. Since the q-dependency of the relaxation modes at these conditions was found to be larger than 2, we have ruled out the possibility of elongated or rod-like structures.

Better solvent conditions should allow for a larger portion of the polymer chains to be unassociated unimers in the solution. In addition, there should be fewer aggregates, which on average contain a smaller amount of polymer chains. Since the aggregates are also expected to be more swollen, the overall size of the aggregates could either increase (due to swelling) or decrease (due to fewer polymer chains in the aggregates). However, swelling of the aggregates is not expected to produce anything of the size scale that would result in the extremely slow relaxation times observed for the third mode. Swelling of the aggregates would also cause an increase of the transmittance [61,75] instead of the observed decline.

If the concentration of unassociated polymer in solution increases at better solvent conditions, depletion flocculation of the aggregates may occur when the critical flocculation concentration of free, unassociated polymer in solution is reached [84]. Depletion flocculation would cause the formation of loose aggregates with an open structure, which is in agreement with the optical clarity of the samples. Depletion flocculation can also explain the formation of very large structures, and the fact that these are promoted at better solvent conditions (low concentrations, smaller amounts of PropOX in the statistical PEtOx-PropOx block, and low temperatures).

As the temperature is raised, the solvent conditions for the temperature sensitive polymers becomes poorer, and the large flocks that gives rise to τ_vs_ disappear. For some of the samples, the values of τ_vs_ decrease when the temperature is raised, suggesting that the flocks become smaller before disappearing completely at even higher temperatures.

## 4. Conclusions

The complex temperature-dependent behavior of three block-copolymers PEtOx_80_-*block*-P(EtOx*_x_*-*stat*-PropOx*_40-x_*) with x = 0, 4, or 8 at different concentrations yields some very interesting results and insights into the process. The correlation functions from dynamic light scattering reveal the presence of four different kinds of structures, of which one to three may be present at the same time. This is summarized in Figure 8, where the temperature regions at which the different relaxation modes are present in the samples are indicated, and the structures causing the different relaxation modes are illustrated. At low temperatures, unimers (R_h,f_) can be observed for all samples. At higher temperatures, the polymers become more hydrophobic leading to enhanced interactions and fewer unassociated unimers. Accordingly, R_h,f_ is no longer evident at higher temperatures. It disappears at lower temperatures when the hydrophobicity of the statistical PEtOx-PropOx block increases (higher amount of PropOx) and at higher concentrations.

R_h,s_ probes aggregates that are present throughout most of the considered temperature region. The sizes of these aggregates increase when the number of polymers they contain is raised, and decreases when the aggregates contract. Since the same forces that results in enhanced aggregation also promotes contraction, the overall sizes of R_h,s_ is a result of these two competing mechanisms. When the temperature is raised, the combined aggregation and contraction of the aggregates causes the samples to become more turbid (reduced transmittance).

At high temperatures, the aggregates (R_h,s_) starts to fragment into smaller micellar-like structures (R_h,x_). These structures are only evident at high temperatures. The fragmentation into smaller units results in a higher transmittance, and the samples become optically clear again. At even higher temperatures, the polymers become even more hydrophobic and re-associate into large aggregates due to collapse of the PEtOx block. At this stage the samples become very turbid (low transmittance), and multiple scattering prevents analyses of the DLS data.

A very slow relaxation mode is present at good solvent conditions, i.e., when the temperature is not too high for the lowest polymer concentration for all polymers, and also at the second lowest concentration for the most hydrophilic polymer. This mode is probably caused by depletion flocculation of aggregates (R_h,s_) when the concentration of free unimers (R_h,f_) becomes high enough.

## Figures and Tables

**Figure 1 polymers-12-01495-f001:**
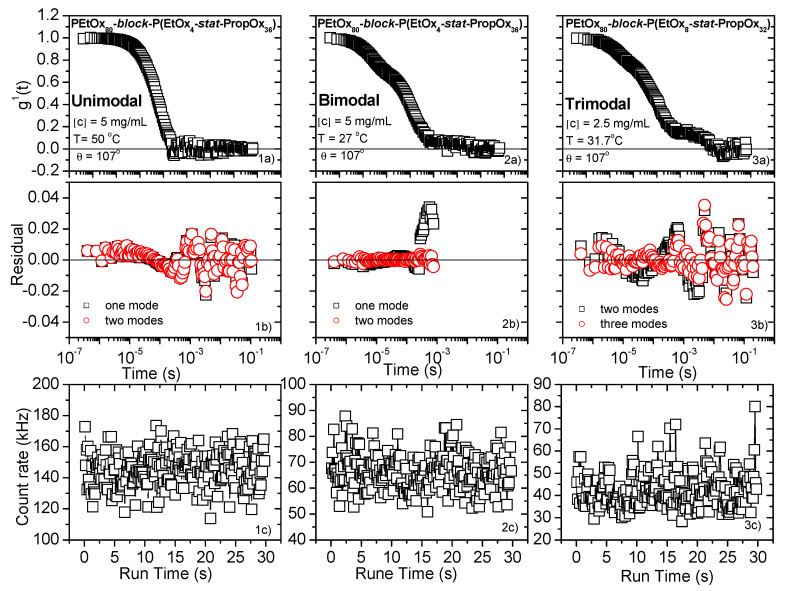
First-order electric field correlation function versus time and residual plots for some representative systems. (**1a**–**c**) For a system that exhibits one relaxation mode. (**2a**–**c**) For a system that displays two relaxation modes and (**3a**–**c**) for a system that shows three relaxation modes. In the latest case, the count rate versus time was plotted to confirm there is no dust in the system and the third modes is a real physical process.

**Figure 2 polymers-12-01495-f002:**
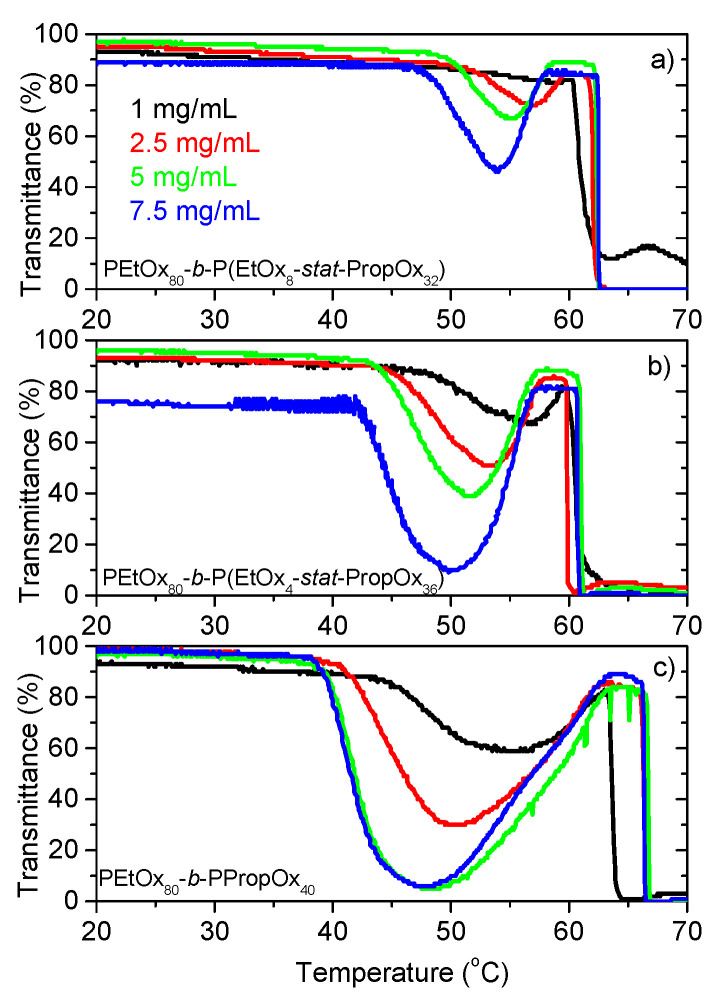
Turbidimetry of the indicated systems. Measured at a heating rate of 1.0 °C/min. (**a**) PEtOx_80_-*block*-P(EtOx_8_-*stat*-PropOx_32_) (**b**) PEtOx_80_-*block*-P(EtOx_4_-*stat*-PropOx_36_) (**c**) PEtOx_80_-*block*-PPropOx_40_.

**Figure 3 polymers-12-01495-f003:**
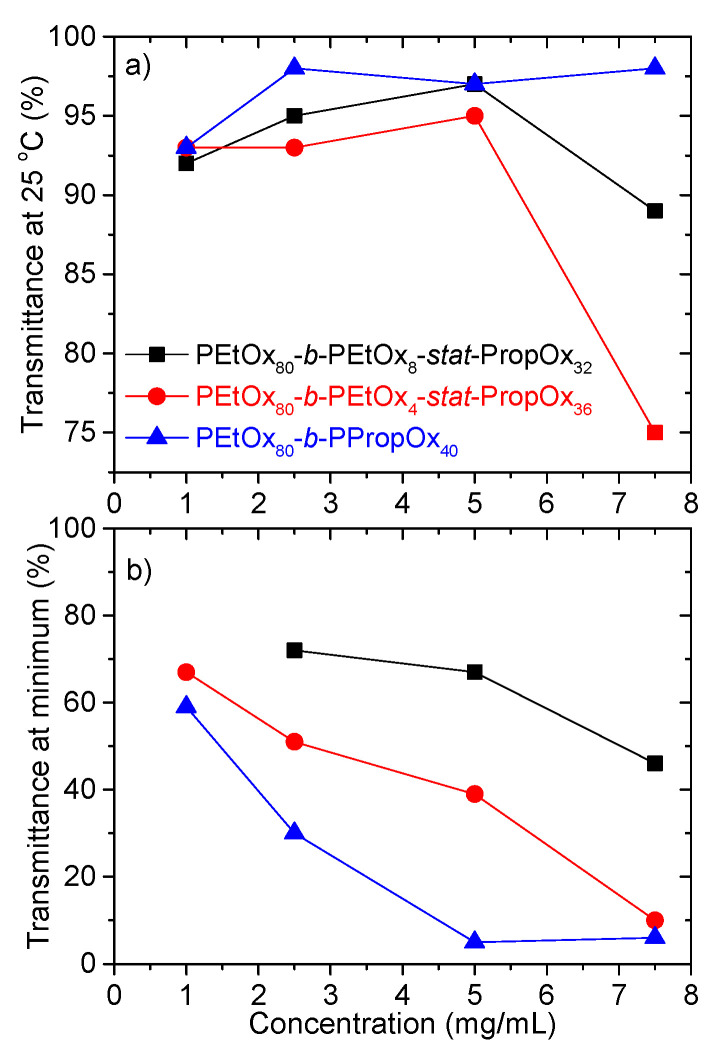
(**a**) Transmittance of the indicated systems at 25 °C. (**b**) Transmittance values at first minimum.

**Figure 4 polymers-12-01495-f004:**
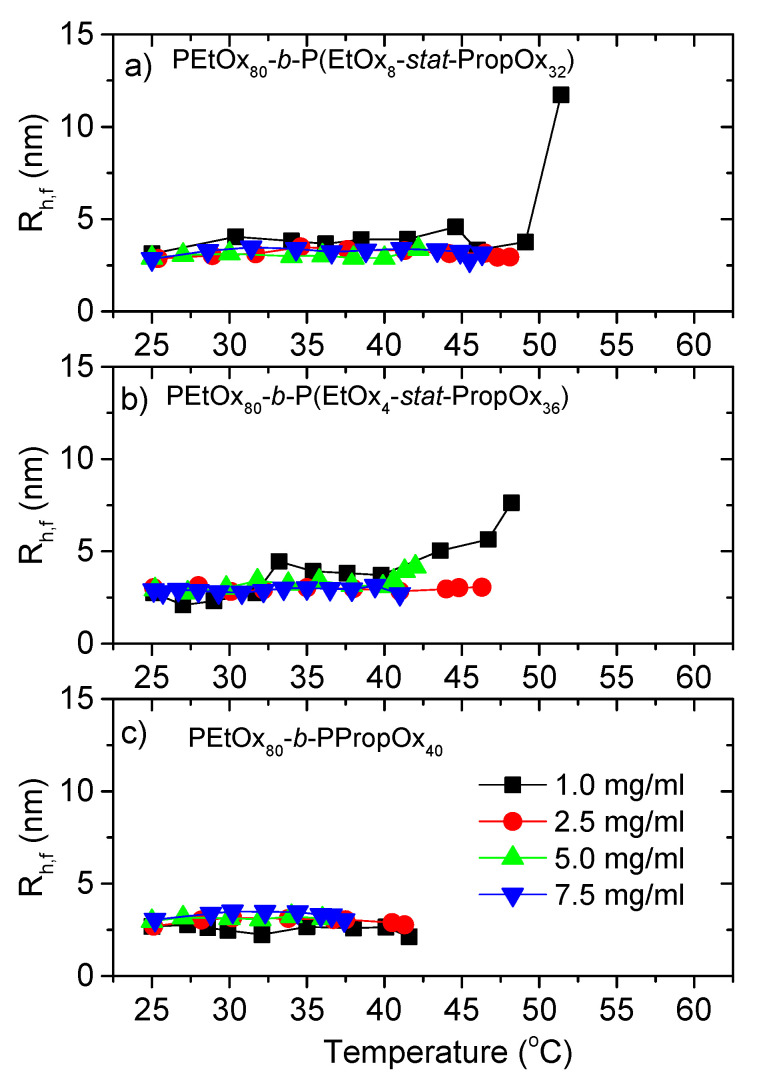
Hydrodynamic radius (R_h,f_) determined from the fast relaxation mode as a function of temperature. Measured with a heating rate of 1 °C/min. The error bars are smaller than the size of the symbols. (**a**) PEtOx_80_-*block*-P(EtOx_8_-*stat*-PropOx_32_) (**b**) PEtOx_80_-*block*-P(EtOx_4_-*stat*-PropOx_36_) (**c**) PEtOx_80_-*block*-PPropOx_40_.

**Figure 5 polymers-12-01495-f005:**
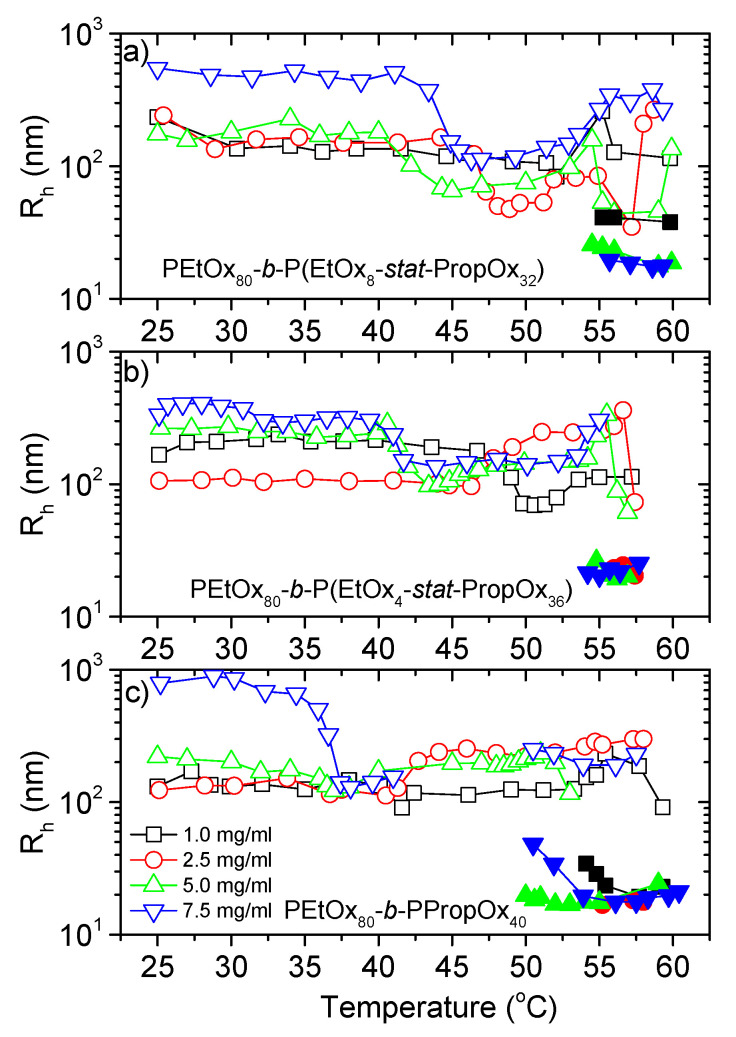
Apparent hydrodynamic radius for the intermediate sized species. R_h,s_ in open symbols, and R_h,x_ in corresponding closed symbols. The missing data between 41 and 50 °C for 7.5 mg/mL PEtOX_80_-*b*-PPropOx_40_ is due to multiple scattering at these conditions, which prevents analysis of the data. Measured with a heating rate of 1 °C/min. The error bars are smaller than the size of the symbols. (**a**) PEtOx_80_-*block*-P(EtOx_8_-*stat*-PropOx_32_) (**b**) PEtOx_80_-*block*-P(EtOx_4_-*stat*-PropOx_36_) (**c**) PEtOx_80_-*block*-PPropOx_40_.

**Figure 6 polymers-12-01495-f006:**
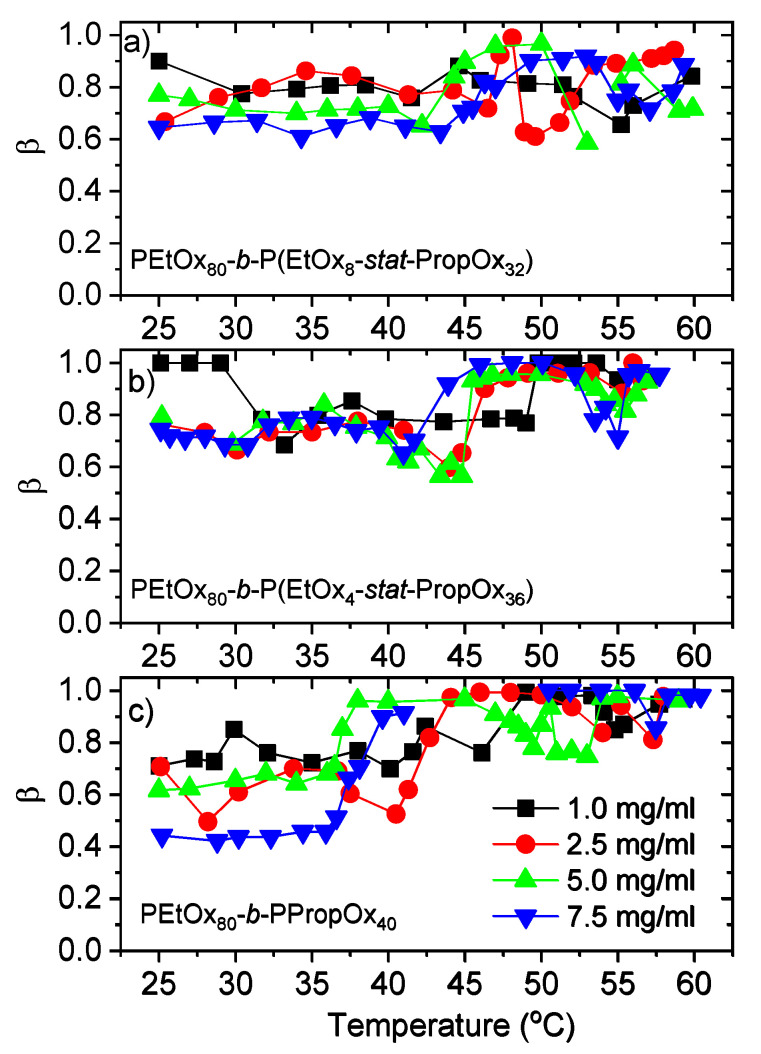
Stretched exponent, β, for the slow mode (R_h,s_). (**a**) PEtOx_80_-*block*-P(EtOx_8_-*stat*-PropOx_32_) (**b**) PEtOx_80_-*block*-P(EtOx_4_-*stat*-PropOx_36_) (**c**) PEtOx_80_-*block*-PPropOx_40_.

**Figure 7 polymers-12-01495-f007:**
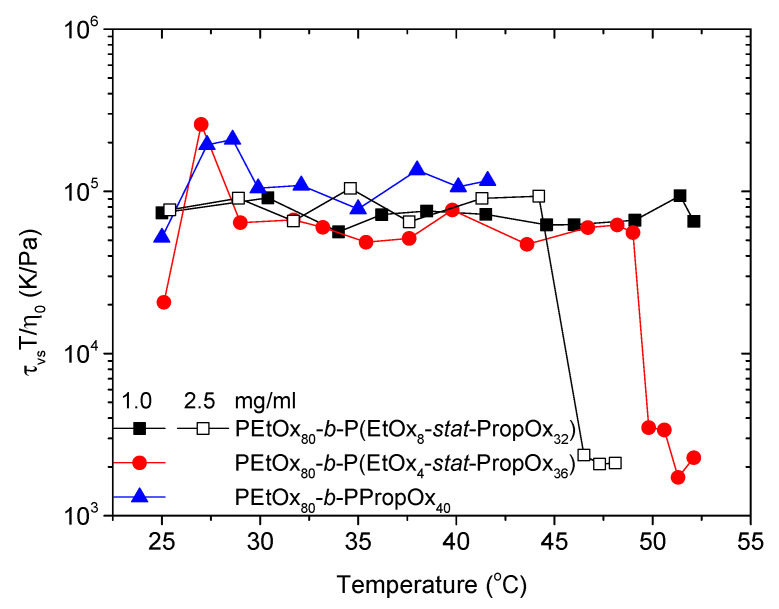
Very slow relaxation time (τ_vs_) of relaxation mode observed at the lowest concentrations. The data has been normalized to compensate for the temperature dependency of the solvent viscosity.

**Figure 8 polymers-12-01495-f008:**
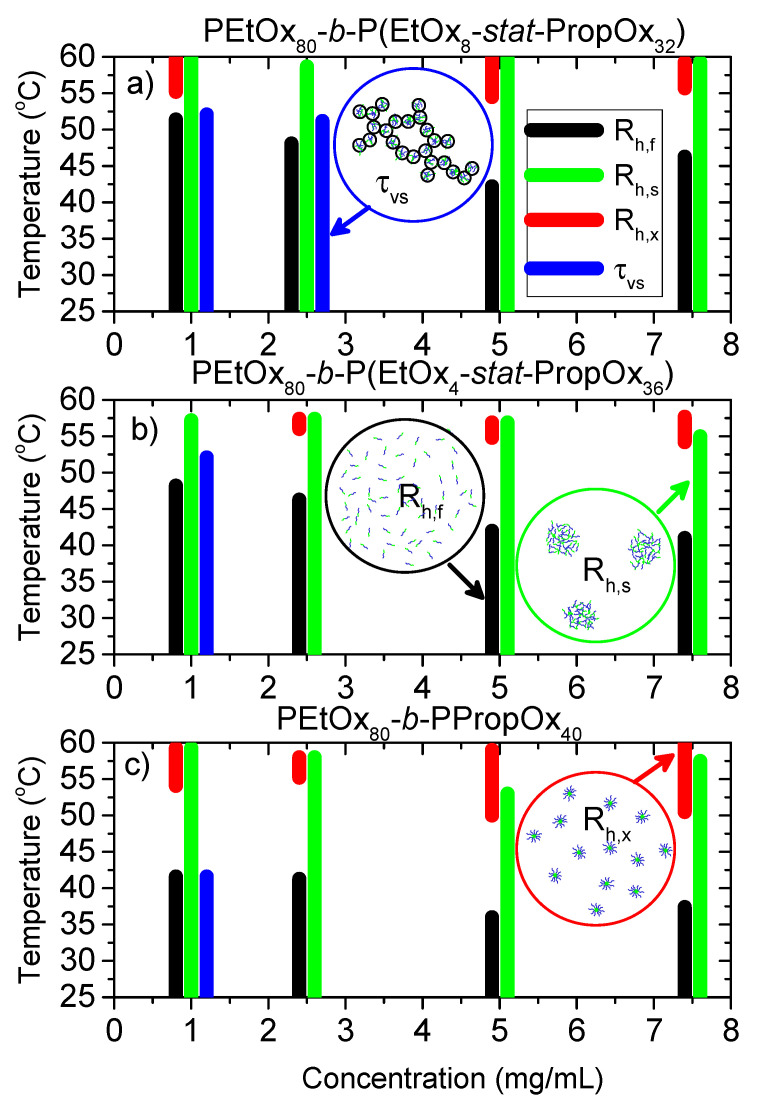
Temperature range in which the different relaxation modes are present, and sketch of the structures that gives rise to the different relaxation modes. R_h,f_—black (2–12 nm), R_h,s_—green (50–900 nm), R_h,x_—red (17–48 nm); τ_vs_—blue (>1 μm). (**a**) PEtOx_80_-*block*-P(EtOx_8_-*stat*-PropOx_32_) (**b**) PEtOx_80_-*block*-P(EtOx_4_-*stat*-PropOx_36_) (**c**) PEtOx_80_-*block*-PPropOx_40_.

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
