# Peer review of "Complex Temperature and Concentration Dependent Self-Assembly of Poly(2-oxazoline) Block Copolymers"

_polymers, 2020, doi:10.3390/polym12071495_

Round 1
Reviewer 1 Report
In this study, the effect of polymer concentration on the temperature-induced self-association of block copolymers containing a poly(2-ethyl-2-oxazoline) block and a random copolymer block consisting of 2-ethyl-2-oxazoline and 2-n-propyl-2-oxazoline (PEtO80-block-P(EtOxx-stat-PropO40-x) was investigated by dynamic light scattering (DLS) and transmittance measurements. The results are helpful for understanding of the self-association mechanism of these block copolymers in solution. This is a well written paper with solid data. I recommend it for publication in Polymers after address the following issues:
- For DLS date, error bar should be provided in Figure 4 and Figure 5 to show the size distribution of micelles.
- In addition to DLS size data, the authors are suggested to provide either TEM or AFM images of the micelles, which can show their morphology and size visually.
Author Response
In this study, the effect of polymer concentration on the temperature-induced self-association of block copolymers containing a poly(2-ethyl-2-oxazoline) block and a random copolymer block consisting of 2-ethyl-2-oxazoline and 2-n-propyl-2-oxazoline (PEtO80-block-P(EtOxx-stat-PropO40-x) was investigated by dynamic light scattering (DLS) and transmittance measurements. The results are helpful for understanding of the self-association mechanism of these block copolymers in solution. This is a well written paper with solid data. I recommend it for publication in Polymers after address the following issues:
- For DLS date, error bar should be provided in Figure 4 and Figure 5 to show the size distribution of micelles.
Answer: The error bars of figure 4 and 5 are smaller than the symbols, and therefore not included. We have added information regarding this in the legend to these figures: "The error bars are smaller than the size of the symbols."
- In addition to DLS size data, the authors are suggested to provide either TEM or AFM images of the micelles, which can show their morphology and size visually.
Answer: Considering that the micelles are very sensitive to both polymer concentration and temperature, drying the samples for TEM or AFM imaging will alter the sizes and morphologies. We have therefore not included such images.
Reviewer 2 Report
The manuscript "Complex Temperature and Concentration dependent Self-assembly of Poly(2-oxazoline) Block Copolymers" by Kjoniksen and coworkers deals with the temperature dependent solution behavior of polyoxazolines. It is a very interesting contribution.
General comments:
1. In my opinion the authors should also show particle size distributions.
2. Can the authors comment on the ratio of the different modes?
3. A similar effect of aggregate formation at ambient temperature of hydrophilic block copolymers was described earlier, which could be helpful to explain the observed behavior:
https://onlinelibrary.wiley.com/doi/abs/10.1002/macp.201700494
https://pubs.acs.org/doi/10.1021/ma300621g
https://www.mdpi.com/2073-4360/9/7/293
Also the recent work by Sawamoto and Terashima could be interesting.
4. How hydrophilic is the copolymer block? Is the copolymer itself water soluble?
5. Although the synthesis of the polymers was described earlier, I would suggest to briefly give some info (Mn, PDI) about the used polymers somewhere.
6. Is there any effect of the preparation procedure on the particle formation? Were the polymer just dissolved in water or was there some additional step involved (solvent exchange, dialysis, ultrasound etc.)?
Specific comments:
7. The authors should check the degree signs in the manuscript and also the punctuation. Sometimes full stops are missing.
8. The referencing style in the manuscript should be checked, sometimes it is superscript.
9. I would suggest to explain the colors in Figure 7 in the caption as well and maybe also address a particle size range to the colors.
10. In the author contributions, some parts of the template are still present.
Author Response
The manuscript "Complex Temperature and Concentration dependent Self-assembly of Poly(2-oxazoline) Block Copolymers" by Kjoniksen and coworkers deals with the temperature dependent solution behavior of polyoxazolines. It is a very interesting contribution.
General comments:
- In my opinion the authors should also show particle size distributions.
Answer: We have included a new Figure 6, showing the b values that describes the width of the size distributions of the samples. Additional text is added to discuss the new figure (lines 307-321):
"The stretched exponent, β, provides information about how broad the size distribution of the aggregates is. b = 1 indicates a monodisperse distribution, and a broader size distribution will give lower values of b. Figure 6 shows b for the slow mode (Rh.s). The other modes exhibited a nearly monodisperse size distribution, with a stretched exponent very close to one (and the exponent was therefore fixed to one to reduce the number of fitting parameters). A narrow size distribution of Rh,f is expected, since the fast mode probes single chains of polymers with a relatively small polydispersity. Rh,x is due to micellar-like structures that is also expected to exhibit a narrow size distribution. The β values for Rh.s varies between 0.4 and 1, and is generally higher (narrower size distributions) at elevated temperatures where large and compact aggregates are formed. At low temperatures, b decreases when the concentration is raised for the most hydrophobic polymer (PEtOx80-b-PPropOx40), and b is also generally lower than for the other polymers in this temperature range. Accordingly, the aggregates formed at low temperatures have a wider size distribution when there are more hydrophobic interactions in the sample."
- Can the authors comment on the ratio of the different modes?
Answer: The amplitudes of the different modes is shown in the attached pdf. However, considering the complex behavior of these samples, including the amplitudes in the manuscript would also result in a rather long and complex discussion regarding of how these are changing with temperature, concentration, and copolymer composition. We think that this will make the manuscript very complex and more difficult to follow, and take the focus away from the main results. We have therefore chosen not to include the amplitudes in the manuscript.
- A similar effect of aggregate formation at ambient temperature of hydrophilic block copolymers was described earlier, which could be helpful to explain the observed behavior:
https://onlinelibrary.wiley.com/doi/abs/10.1002/macp.201700494
https://pubs.acs.org/doi/10.1021/ma300621g
https://www.mdpi.com/2073-4360/9/7/293
Also the recent work by Sawamoto and Terashima could be interesting.
Answer: Thank you for the suggestion. We have included references to these papers together with some additional text (line 255-258):
" This is not uncommon for thermoresponsive polymers [73, 77-79], and even double hydrophilic block copolymers has been observed to form aggregates in dilute aqueous solutions [80-82]. The latter was explained by differences in hydrophilicities between the two blocks, resulting in different water uptake [80]. It is reasonable to assume that similar effects influence the aggregation behavior of thermoresponsive copolymers below the LCST."
- How hydrophilic is the copolymer block? Is the copolymer itself water soluble?
Answer: As shown by the DLS results at low temperature, copolymers are mostly present as unimolecular individual chains indicating that the copolymer block is water soluble at low temperatures, but not at high temperatures.
- Although the synthesis of the polymers was described earlier, I would suggest to briefly give some info (Mn, PDI) about the used polymers somewhere.
Answer: We have added information regarding Mn and PDI of the polymers (line 118-120):
"PEtOx80-block-PPropOx40 has Mn = 8700 g/mol and PDI = 1.22; PEtOx80-block-P(EtOx4-stat-PropOx36) has Mn = 8200 g/mol and PDI = 1.23; PEtOx80-block-P(EtOx8-stat-PropOx32) has Mn = 9200 g/mol and PDI = 1.20."
- Is there any effect of the preparation procedure on the particle formation? Were the polymer just dissolved in water or was there some additional step involved (solvent exchange, dialysis, ultrasound etc.)?
Answer: The polymers were just dissolved in water without any additional steps. We have not tried other preparation procedures. We have clarified the preparation procedure (line 120-122):
"Solutions of the polymers were prepared in deionized water, and were stirred at room temperature until all polymer was dissolved or dispersed."
Specific comments:
- The authors should check the degree signs in the manuscript and also the punctuation. Sometimes full stops are missing.
Answer: This has been corrected.
- The referencing style in the manuscript should be checked, sometimes it is superscript.
Answer: This has been corrected.
- I would suggest to explain the colors in Figure 7 in the caption as well and maybe also address a particle size range to the colors.
Answer: We have added the requested information in the legend to Figure 7:
"Rh,f - black (2-12 nm), Rh,s - green (50-900 nm), Rh,x - red (17-48 nm); tvs - blue (> 1 mm)."
- In the author contributions, some parts of the template are still present.
Answer: This has been corrected.
